# Strength and Durability Properties of Antimony Tailing Coarse Aggregate (ATCA) Concrete

**DOI:** 10.3390/ma14195606

**Published:** 2021-09-27

**Authors:** Long Li, Jianqun Wang, Longwei Zhang, Renjian Deng, Saijun Zhou, Gongxun Wang

**Affiliations:** Hunan Provincial Key Laboratory of Structures for Wind Resistance and Vibration Control, School of Civil Engineering, Hunan University of Science and Technology, Xiangtan 411201, China; lilong15173271886@163.com (L.L.); 800912deng@sina.com (R.D.); sjzhou@hnust.edu.cn (S.Z.); wanggx@hnust.edu.cn (G.W.)

**Keywords:** antimony tailing waste rock (ATWR), antimony tailing coarse aggregate (ATCA), natural coarse aggregate (NCA), construction performance, mechanical property, long-term behavior, durability performance

## Abstract

Antimony (Sb) is a trace element applied widely in modern industry. A large number of tailing solid wastes are left and accumulated in the mining area after purifying the precious antimony from the antimony ores, causing serious pollution to the environment. The major aim of this study is to investigate the feasibility of utilizing antimony tailing coarse aggregate (ATCA) as a complete substitute for natural coarse aggregate (NCA) in high-strength concrete. Concrete specimens with 25%, 50%, 75%, and 100% ATCA replacing the NCA in conventional concrete were prepared for evaluating the performance of ATCA concrete. The investigators find that ATCA concrete has good workability, and the mechanical properties and long-term behavior (shrinkage and creep) of ATCA concrete with all replacement levels are superior to those of NCA concrete. The durability indices of ATCA concrete, such as the frost-resistant, chloride permeability, and resistance to carbonation, are better than those of NCA concrete. While the alkali activity and cracking sensitivity behavior of ATCA concrete seem to be decreased, nevertheless, the difference is not significant and can be neglected. The researchers demonstrate that all of the control indices of ATCA concrete meet the requirements of the current industry standards of China. Overall, ATCA can be used in concrete to minimize environmental problems and natural resources depletion.

## 1. Introduction

Concrete is one of the most popularly used building and construction materials and has been widely researched [1,2,3]. In recent years, the green and sustainable production of concrete has received much attention [4,5,6]. Coarse aggregate (CA), as the largest component of the concrete mix ratio, has continued to be increased in recent years. Therefore, a large number of quarries are mined, negatively impacting the environment. The traditional ways to produce CA are not sustainable. Therefore, researchers are actively searching for substitutes for aggregates with minimum or no negative impact on the natural resources [7,8]. Some new eco-friendly concrete, including tailing concrete, recycled aggregate concrete, and waste ceramic aggregate concrete, have been extensively researched and successfully applied in civil engineering [9].

China is the largest consumer and producer of concrete in the world. Meanwhile, China has the richest antimony deposits in the world and produces millions of tons of antimony ore tailing. Since the beginning of the 20th century, China has become the world’s largest producer of antimony and its compounds [10]. A large number of tailing solid wastes are accumulated in the mining area after purifying the precious antimony from the antimony ores. To dispose these solid wastes, it is necessary to build large tailing dams and take strict isolation measures, which require the use of additional land and cost [11].

Metal tailing has long been considered a worldwide environmental problem for the earth and aquatic pollution [12,13]. These solid wastes release harmful elements under the action of alternation of dry and wet conditions, high and low temperatures, and even freeze and thaw conditions. These hazard elements enter into the soil and surrounding rivers via the runoff and infiltration of rainwater [14,15], which not only cause pollution to the surrounding environment of the mining area but also move to other living areas [16]. In order to deal with these solid waste materials, extensive research has been conducted around the world, and some treatment methods have been developed. A feasible method is to utilize the metal tailing in concrete as a partial or complete substitution for aggregate. In this way, low-cost green concrete is obtained, and meanwhile, harmful substances can be solidified in the concrete [17,18].

Onuaguluchi and Eren [19,20] investigated the feasibility of using copper tailing as a partial substitute of cement for mortars material. It was demonstrated that the copper tailing led to an improvement in mechanical strength, abrasion resistance, and resistance to chloride penetration. Blessen et al. [21] conducted an experiment on concrete containing copper tailing as fine aggregate, which indicated that the maximum replacement amount was 60%. Sharma et al. [22] reported the reuse of copper tailing as a substitute for natural sand in self-compacting concrete. It was shown that, compared with controlled samples, even with 100% copper slag reused as a substitute for natural sand, the concrete possessed a better mechanical strength and absorption properties. Huang et al. [23] utilized copper tailing in autoclaved aerated concrete and proposed an effective method to decrease the CO_2_ emission in the concrete production process.

Zhao et al. [24] studied different substitute levels of iron ore tailing (IOT) for natural sand in ultra-high-performance concrete. Test results showed that the mechanical behavior of concrete with no more than 40% natural sand replaced was nearly the same with the controlled sample. Shettima et al. [25] evaluated the performance of concrete with different substitute amount of IOT as natural sand. Test results indicated that IOT had little negative effect on the construction performance, while mechanical strength was superior to the control specimen. Lv et al. [26] carried out a series of experiments to investigate the behavior of dam concrete with IOT as natural aggregate. It was demonstrated that the tailing aggregate concrete could possess a comparable mechanical behavior and frost-resistant performance with superior thermal performance. Liu et al. [27] studied the sprayed concrete containing IOT substituted for the natural sand and suggested the best substitution was 20%. IOT was also utilized in green engineered cementitious composites [28], metal tailing porous concrete [29], and steam-cured precast concrete [30]. In addition, tungsten ore tailing [31], gold ore tailing [32], zinc ore tailing [33], and molybdenum ore tailing [34] were also reported for reusing in concrete. Wang investigated the feasibility of utilizing antimony tailing waste rock (ATWR) for a practical project, providing a new way for utilizing ATWRs [35]. It is demonstrated that all of the quality control indices of antimony tailing coarse aggregate (ATCA) meet the requirements of the current industry standards of China. 

As mentioned above, although the literature on other metal ore tailing utilizing in concrete is relatively rich, little investigation on the utilization of antimony ore tailing as CA has been reported. The goal of this research is to study the feasibility of ATCA as a complete substitute for natural coarse aggregate (NCA). The workability, mechanical property, long-term performance, and durability of ATCA concrete and NCA concrete are studied and compared in the present work.

The rest of this paper is organized as follows. Section 2 introduces the materials and mix ratio of ATCA concrete. Section 3 describes the workability of fresh ATCA concrete, such as slump, air content, setting time, and apparent density and so on. Section 4 illustrates the mechanical property of ATCA concrete, including cubic compressive strength, splitting tensile strength, prism compressive strength, and elasticity modulus. Section 5 shows the long-term behavior of ATCA concrete, including drying shrinkage and creep. Section 6 researches the durability performance of ATCA concrete, involving freezing and thawing, alkali activity, chloride ion penetration, cracking sensitivity, and carbonation. Finally, Section 7 concludes the article and gives some recommendations for future work.

## 2. Materials and Mix Ratio of Concrete

### 2.1. Materials

#### 2.1.1. Cement

In this investigation, Ordinary Portland cement with a strength class of 42.5, in accordance with Chinese Standard GB 175-2007 was used [36]. For the cement, the specific gravity was 3130 kg/m^3^, the specific surface area was 312 m^2^/kg, the normal consistency was 27.7%, and the initial and final setting time was 168 min and 207 min, respectively. The compressive strength of cement mortar at 3 days, 7 days, and 28 days iwas 26.7 MPa, 38.5 MPa and 52.8 MPa, respectively. The main chemical composition of cement is shown in Table 1.

#### 2.1.2. Aggregate

The ATWR was obtained by 5-point sampling method in the antimony waste ore stacking area in Xikuangshan (XKS). The XKS Sb mine, located near Lenshuijiang City, Hunan Province, China, is the largest Sb mines in the world [37,38,39]. Due to Sb mining and smelting processes, a large number of tailing solid wastes are left and accumulated in the mining area [40,41]. During the field investigation, the authors found that there were millions of tons of ATWRs in XKS (Figure 1). These ATWRs occupy valuable land resources, resulting in serious Sb contamination of the local environments [42,43]. The sampled ATWR was mixed and crushed into ATCA, of which the particle size was 5 mm–20 mm. 

### 2.2. Mix Ratio of Concrete

The concrete mix proportion was determined through multiple trial mixes with the absolute volume method. Poly-carboxylate superplasticizer was added to concrete with 0.2% by the weight of cementitious materials. Table 2 shows the five batches of mixture proportions. T0 was the control mix with NCA. While in the other 4 mixes, T25, T50, T75, T100, the NCA in the samples was substituted by ATCA, with different levels of 25%, 50%, 75%, and 100%, respectively.

## 3. The Workability of Fresh Concrete

### 3.1. Experiment Methodology

The workability of fresh concrete is a key factor in assessing whether it can be applied to construction projects. CA is the largest component of the concrete mix ratio, playing an important role as the skeleton. The shape and particle size of CA will affect the construction performance of concrete. The slump, air content, setting time, and apparent density was tested based on the Chinese Standard GB/T 50080-2016 [44]. The workability of fresh concrete is illustrated and assessed in the following subsection.

### 3.2. Results and Discussion

The workability of fresh concrete, such as the slump, setting time, air content, and apparent density, is listed in Table 3. It is demonstrated that all indexes meet specification requirements [44,45,46]. The apparent density of ATCA concrete was slightly higher than that of NCA concrete, and the value was only 1%. Other indices decrease with the increasing replacement amount of ATCA. Nevertheless, it was explicitly found that the difference in workability between ATCA and NCA concrete was very small. It can be inferred that the influence of CA on concrete construction performance was negligible.

## 4. Mechanical Property Experiments

### 4.1. Specimen Design

It is important to understand the development of strength, especially at the early age of concrete structures [47,48]. The mechanical properties, such as cubic compressive strength, splitting tensile strength, prism compressive strength, and elasticity modulus, were conducted to investigate the ATCA as a complete substitute for NCA in high strength concrete in this study. The early age and long-term mechanical properties were tested at the curing age of 3, 7, 28, 90, and 180 days. The test methods were based on Chinese Standard GB/T 50081-2019 [49]. The dimensions of the specimens used for the experimental study are listed in Table 4.

### 4.2. Results and Discussion

#### 4.2.1. Compressive Strength of Cubic Sample

As illustrated in Figure 2, it can be seen that the difference is very small during the first 3 days. However, the measured values of compressive strength of T25, T50, T75, T100 at the 28th day were 58.9 MPa, 61.5 MPa, 63.5 MPa, and 64.5 MPa, which were 102%, 108%, 110%, and 112% of the controlled sample (T0), respectively. Moreover, when the curing age was 180 days, the results of compressive strength of T25, T50, T75, T100 were 102%, 107%, 110%, and 116% of the controlled sample, respectively. Therefore, it can be considered that the compressive strength of ATCA concrete at all replacement levels was superior to that of the NCA concrete at all curing ages. Furthermore, the compressive strength of the concrete increases with the increasing replacement level of ATCA.

#### 4.2.2. Splitting Tensile Strength of Cubic Sample

The tensile strength of concrete is a vital factor in evaluating the crack resistance of structures [50]. The splitting tensile strength of the cubic sample was tested simultaneously with the compressive strength. The results are shown in Figure 3. The measured values of splitting tensile strength of T25, T50, T75, T100 at 28 days are 3.92 MPa, 4.01 MPa, 4.11 MPa, and 4.17 MPa, which were 101%, 103%, 106%, and 107% of the controlled sample, respectively. At 180 days, the tested results of splitting tensile strength of T25, T50, T75, T100 were 103%, 106% 108%, and 110% of the control. The development tendency of splitting tensile strength was consistent with the compressive strength of ATCA concrete.

#### 4.2.3. Compressive Strength and Compressive Elastic Modulus of Prism Specimens

The test result of prism compressive strength and elastic modulus for NCA concrete and ATCA concrete are shown in Figure 4. The values of prism compressive strength of T25, T50, T75, T100 at 28 days were 44.6 MPa, 45.8 MPa, 47.4 MPa, and 48.5 MPa. The percentage increase of prism compressive strength, compared with the control sample, for T25, T50, T75, and T100 was 2%, 4%, 8%, and 10%, respectively. While for the age of 180 days, the percentage increase for T25, T50, T75, and T100 was 3%, 6%, 10%, and 13%, respectively.

As can be seen, from Figure 4b, the tested results of compressive elastic modulus of T25, T50, T75, and T100 at 28 days were 3.88 × 10^4^ MPa, 3.94 × 10^4^ MPa, 4.04 × 10^4^ MPa, and 4.11 × 10^4^ MPa, respectively. Compared with T0, the percentage increase was 3%, 4%, 7%, and 9%, respectively. While for the long-term elastic modulus, when the curing age was 180 days, the percentage increase was 3%, 6%, 9%, and 11%, respectively. Since compressive strength is an adequate index for mechanical properties, a close relationship exists between compressive strength and elastic modulus of ATAC concrete.

In this section, the mechanical behaviors such as cube compressive strength, splitting tensile strength, prism compressive strength, and elastic modulus of all the samples were tested simultaneously. The results show that the mechanical behavior of ATCA concrete with all replacement levels were superior to NCA concrete. Furthermore, at a certain curing age, the mechanical behavior of concrete containing ATCA is enhanced with the increased replacement level. This might be due to the superior mechanical property of ATCA, and better bond between cement and ATCA for the irregular surface. In related research, Shettima et al. [25] reported a similar trend when IOT was reused as an aggregate. They observed that the compressive strength and elastic modulus were superior to the control sample (conventional concrete without IOT) at all levels of replacement. It was demonstrated that the ATCA as a complete substitution for NCA possesses perfect physical and mechanical properties. In order to further study the feasibility of ATCA for concrete, the long-term performance was studied in the next section.

## 5. Long-Term Behavior Experiments

### 5.1. Specimen Design

The shrinkage of concrete is one of the main causes of structural cracking [51]. The CA can restrain the shrinkage of concrete [52,53]. Creep is one of the important indices for the long-term performance of concrete. While there appears to be little research on the creep behavior of tailing concrete. The creep performance of ATCA concrete will provide some references for future research.

The drying shrinkage and creep were tested simultaneously in the laboratory with constant temperature and humidity based on the Chinese Standard GB/T 50082-2009 [54]. The dimensions of the specimens used for the experimental study are listed in Table 5.

### 5.2. Results and Discussion

#### 5.2.1. Drying Shrinkage

The test results of drying shrinkage are shown in Figure 5. Throughout the period of the experiment, the shrinkage of concrete with all replacement levels of ATCA was less than that of the controlled sample. At an early age, the shrinkage strains of all samples were almost the same. However, at the curing age of 180 days, the shrinkage strains of T25, T50, T75, and T100 were, respectively, 301 × 10^−6^, 309 × 10^−6^, 315 × 10^−6^, 318 × 10^−6^, and 329 × 10^−6^, which were, respectively, 3%, 4%, 6%, and 9% less than those of the controlled sample.

#### 5.2.2. Creep

The creep tests of concrete specimens with loading ages of 7 days and 28 days were conducted, and the creep coefficients were used to denote the creep performance. The test results are shown in Figure 6. It can be observed from the figure that the creep of concrete with all replacement levels of ATCA was less than the controlled sample. At an early age, the creep coefficients of all samples were almost the same. However, at the curing age of 180 days, the creep coefficients of T25, T50, T75, and T100 loaded at the age of 7 days were respectively 1.99, 1.93, 1.91, and 1.85, which were respectively 3%, 6%, 7%, 10% less than the controlled sample. Furthermore, the creep coefficients decreased with the increase of the loading age.

In this section, the drying shrinkage and creep of all the samples were tested at the same time. The results showed the drying shrinkage and creep of ATCA concrete with all replacement levels were superior to those of the controlled sample. Furthermore, the drying shrinkage and creep of ATCA concrete decreased with the increased replacement level of ATCA. The factor for lower drying shrinkage and creep of ATCA concrete could be attributed to the higher compressive strength of ATCA, and better bond between cement and ATCA. To further study the feasibility of ATCA for concrete, the durability behavior is discussed in the next section.

## 6. Durability Performance Experiments

### 6.1. Specimen Design

Durability is the most critical factor affecting the life cycle of concrete structures. Some indices such as freezing and thawing, carbonation, and chloride ion penetration have been investigated in the related studies. Since there may be some impurities in the tailing aggregates, the cracking sensitivity and alkali activity should be paid attention to as well. In this section, the durability performance such as freezing and thawing, alkali activity, chloride ion penetration, cracking sensitivity, and carbonation was studied systematically. The dimensions of the specimens used for the experimental study are listed in Table 6.

### 6.2. Results and Discussion

#### 6.2.1. Frost-Resistant Performance

The frost resistance ability of concrete is expressed by surface peeling and relative dynamic modulus of elasticity, which is measured by the ultrasonic method [55]. The test result can help to determine the durability of concrete under extreme low-temperature conditions. The results of freeze-thaw test are shown in Figure 7. The frost-resistant performance of all samples meets the index of F300. After 300 cycles of freeze-thaw, the results of mass loss of T0, T25, T50, T75, and T100 were, respectively, 2.07%, 2.02%, 1.99%, 1.92%, and 1.90%, which were far lower than 5% required by the code. The values of relative dynamic elastic modulus of T0, T25, T50, T75, and T100 were, respectively, 80.1%, 81.9%, 83.7, 84.9%, and 85.8%, which were far higher than 60% required by the code. It can be seen from the test results that the frost-resistant performance of concrete with ATCA was slightly better. The factor for better frost-resistant performance of ATCA concrete resulted from the superior water absorption and firmness of ATCA [35].

#### 6.2.2. Alkali Activity

Under complex conditions, certain minerals in the aggregate could react chemically with the alkali (K_2_O, Na_2_O) in the concrete, causing expansion, crack, or even damage in concrete. In order to evaluate the alkali-silicic acid reaction of ATCA aggregates, the alkali activity experiment was conducted. For 14 days, all samples were placed in the NaOH solution, of which the concentration was 1 mol/L, and the temperature was (80 ± 2) °C. The length of the sample was measured, by which the expansion rate of length was calculated, and the alkali activity of aggregate was evaluated.

The experimental results are shown in Figure 8. After the 14-day immersion in alkali solution, there was no crack, swell, colloid overflow, or other undesirable phenomena. It was explicitly found that the expansion rate was stable at 10 days. The expansion rates of T0, T25, T50, T75, and T100 were 0.71‰, 0.73‰, 0.77‰, 0.79‰, and 0.80‰, respectively, which were all less than 1‰ of the standard required. The alkali activity of samples seemed to increase with the substitution amount of ATCA. However, the difference was not significant and can be neglected.

#### 6.2.3. Chloride Permeability

It is well known that chloride migrates with moisture in concrete [56,57], and it is important to monitor chloride permeability and moisture level in a concrete structure [58,59]. In this research, the chloride permeability of concrete was determined by the electric flux method illustrated as Figure 9. The results of the measured electric flux of T0, T25, T50, T75, and T100 were 869 coulombs, 862 coulombs, 853 coulombs, 843 coulombs, and 832 coulombs after 28 days of curing, respectively. The difference in chloride permeability of concrete was not significant. Based on the code, if the passed charge was above 4000 coulombs, chloride ion penetrability was high penetrability (grade I), 2000–4000 coulombs was moderate penetrability (grade II), 1000–2000 coulombs were low penetrability (grade III), and 100–1000 coulombs was very low penetrability (grade IV). According to the evaluation index, the chloride permeability of all samples was very low (grade IV).

#### 6.2.4. Cracking Sensitivity

High-strength concrete structures are more prone to cracking [60], and cracks in concrete enable water migration [61]. Therefore, crack sensitivity and detection have received much attention [62,63]. In this research, the cracking sensitivity of concrete specimens at an early age was tested with the plate method (Figure 10). The values of measured cracking area per unit area of T0, T25, T50, T75, and T100 concrete were 239 mm^2^/m^2^, 243 mm^2^/m^2^, 248 mm^2^/m^2^, 250 mm^2^/m^2^, and 256 mm^2^/m^2^, respectively. As per the code, if the measured cracking area per unit area was above 1000 mm^2^/m^2^, the cracking sensitivity of concrete was classified as grade L-I, 700–1000 mm^2^/m^2^ was classified as grade L-II, 400–700 mm^2^/m^2^ was classified as grade L-III, 100–400 mm^2^/m^2^ was classified as grade L-IV and less than 100 mm^2^/m^2^ was classified as grade L-V. According to the evaluation index, the cracking sensitivities of all samples were very good (grade L-IV).

#### 6.2.5. Carbonation Depth

The test result of carbonation depth is shown in Figure 11. The measured values of carbonation depth of T25, T50, T75, and T100 were 5.30 mm, 5.61 mm, 5.82 mm, 6.03 mm, and 6.15 mm, which were 6%, 10%, 14%, 16% less than that of the controlled sample. As per the code, if the carbonation depth was above 30 mm, concrete resistance to carbonation was classified as grade I, 20–30 mm was classified as grade II, 10–20 mm was classified as grade III, 0.1–10 mm was classified as grade IV and less than 0.1 mm was classified as grade V. According to the evaluation index, the resistance abilities to carbonation of all samples were very good (grade IV). Furthermore, with the increase of substitution level, the ability of resistance to carbonation was enhanced.

## 7. Conclusions

Based on the systematical studying of the performance of antimony tailing coarse aggregate (ATCA) concrete and the comparison with natural coarse aggregate (NCA) concrete, the researchers have the following main conclusions:(1)ATCA as a complete substitution for NCA in concrete possesses good workability, which is nearly the same as NCA concrete.(2)ATCA as a complete substitution for NCA in concrete possesses perfect physical and mechanical properties. The cubic compressive strength, cubic splitting tensile strength, prism compressive strength, and elastic modulus of ATCA concrete with all replacement levels is superior to that of NCA concrete.(3)The drying shrinkage and creep of ATCA concrete decrease with the increased replacement level of ATCA. Furthermore, the drying shrinkage and creep of ATCA concrete with all replacement levels are superior to the controlled sample.(4)The ability of resistance to carbonation, the frost-resistant, and the chloride permeability performance of ATCA concrete are enhanced. While the alkali activity and cracking sensitivity behavior of the samples with ATCA seem to be decreased. Nevertheless, the difference is not significant and can be neglected.(5)It is found that all of the control indices meet the requirements of the current industry standards of China. The utilization of ATCA as a complete substitute for NCA in high-strength concrete has very important environmental and economic benefits.

## Figures and Tables

**Figure 1 materials-14-05606-f001:**
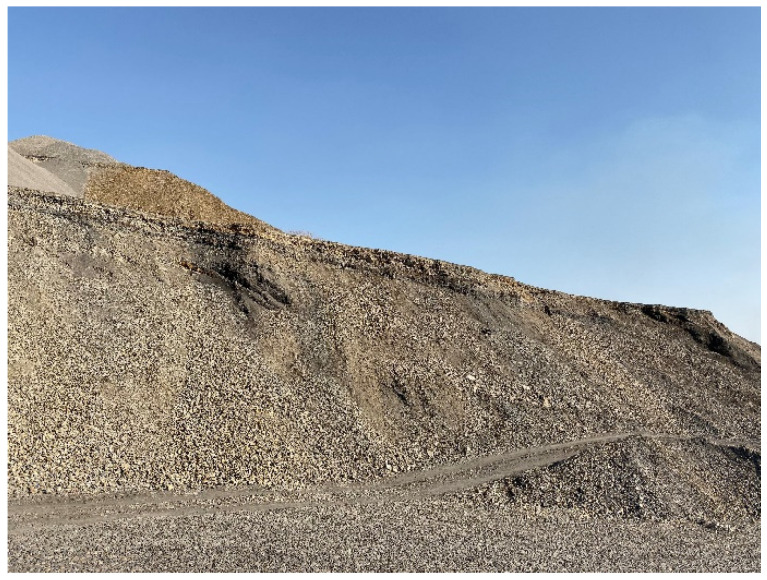
The ATWRs in Xikuangshan (XKS) Sb mine.

**Figure 2 materials-14-05606-f002:**
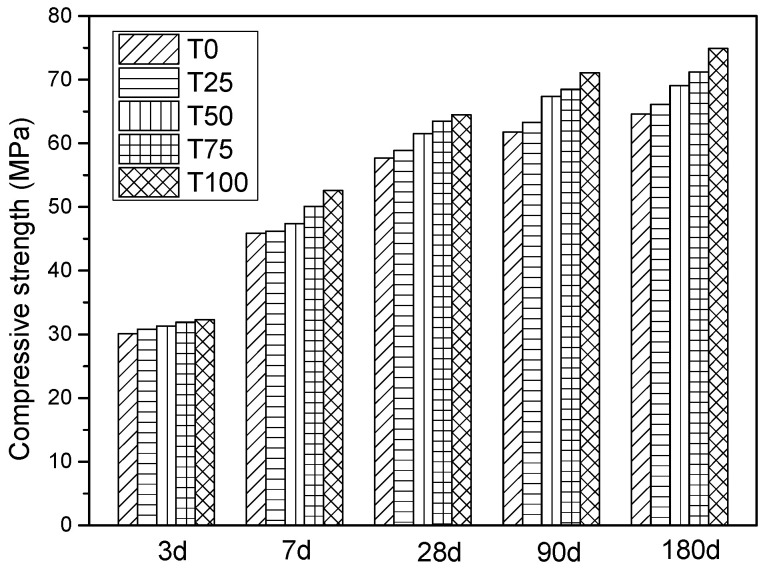
The test results of compressive strength of cubic sample.

**Figure 3 materials-14-05606-f003:**
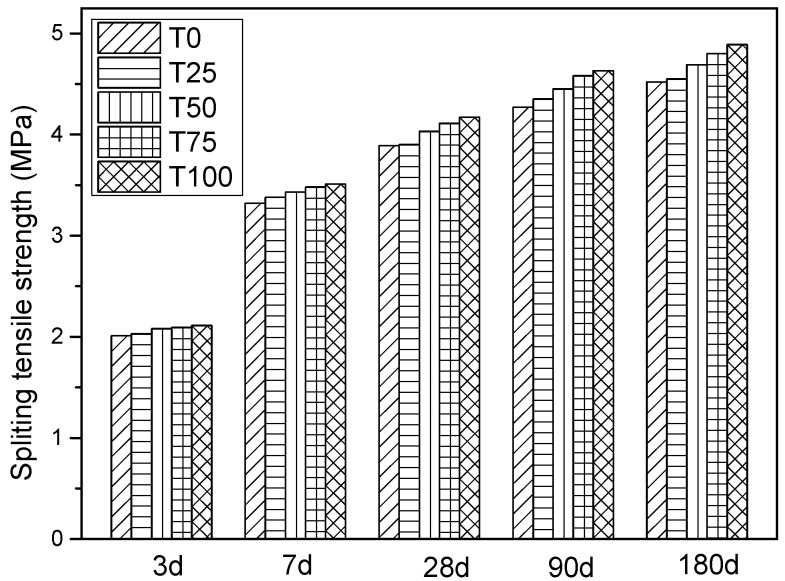
The test result of splitting tensile strength.

**Figure 4 materials-14-05606-f004:**
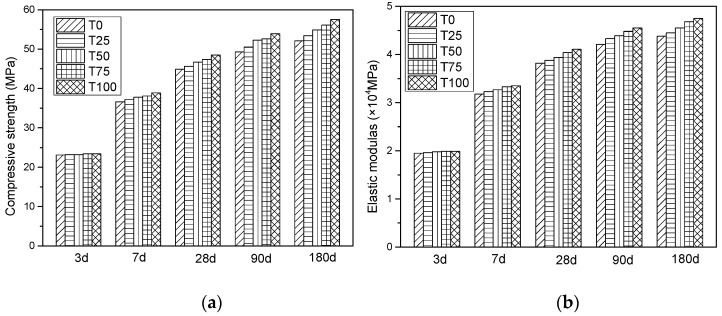
The test result of prism compressive strength and elastic modulus: (**a**) compressive strength; (**b**) compressive elastic modulus.

**Figure 5 materials-14-05606-f005:**
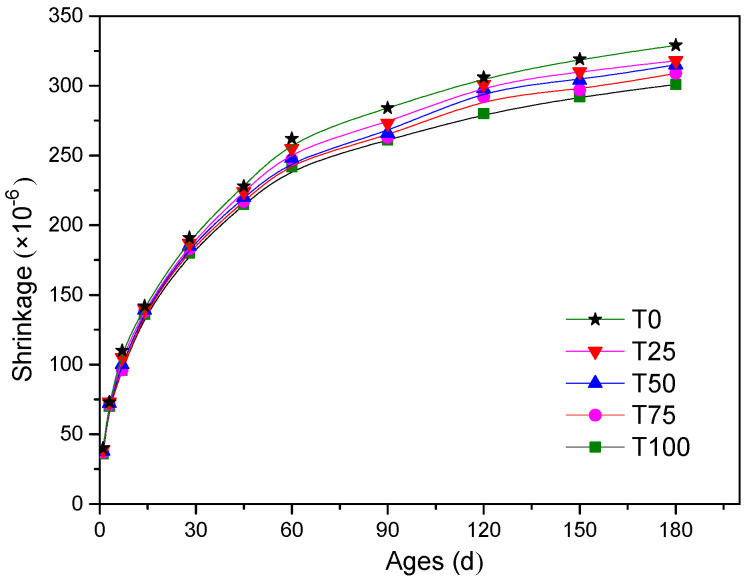
Comparison of drying shrinkage.

**Figure 6 materials-14-05606-f006:**
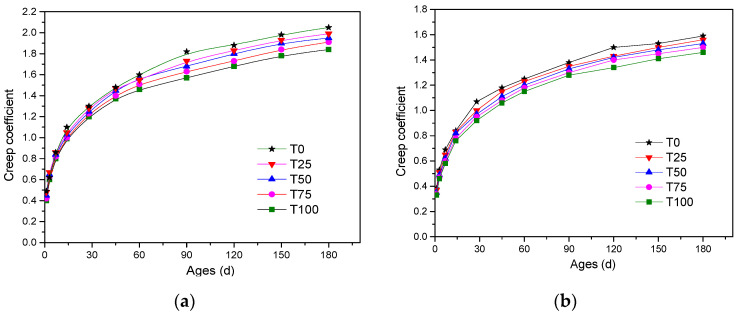
Comparison of creep: (**a**) creep for concrete loaded at the age of 7 days; (**b**) creep for concrete loaded at the age of 28 days.

**Figure 7 materials-14-05606-f007:**
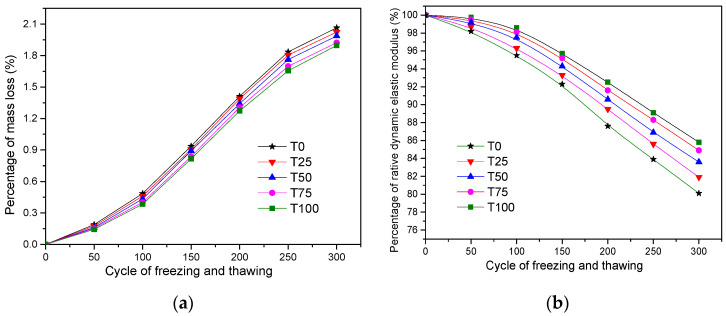
Comparison of frost-resistant performance: (**a**) the test result of mass loss; (**b**) the test result of relative dynamic elastic modulus.

**Figure 8 materials-14-05606-f008:**
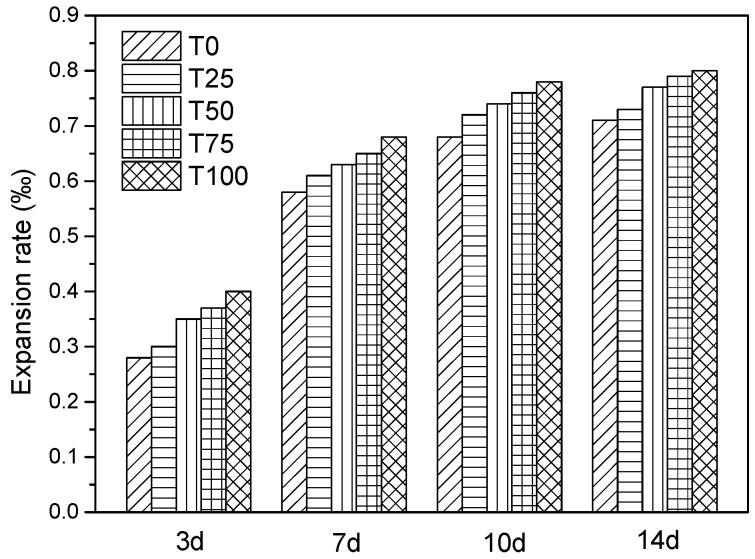
Comparison of alkali activity.

**Figure 9 materials-14-05606-f009:**
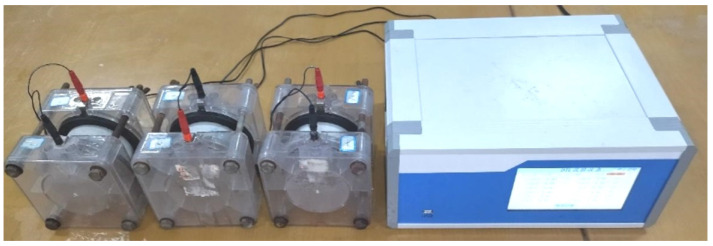
Chloride permeability experiment.

**Figure 10 materials-14-05606-f010:**
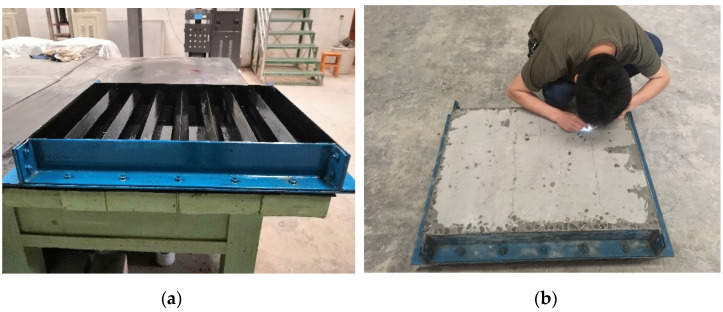
Cracking sensitivity experiment: (**a**) blade mold; (**b**) crack observation.

**Figure 11 materials-14-05606-f011:**
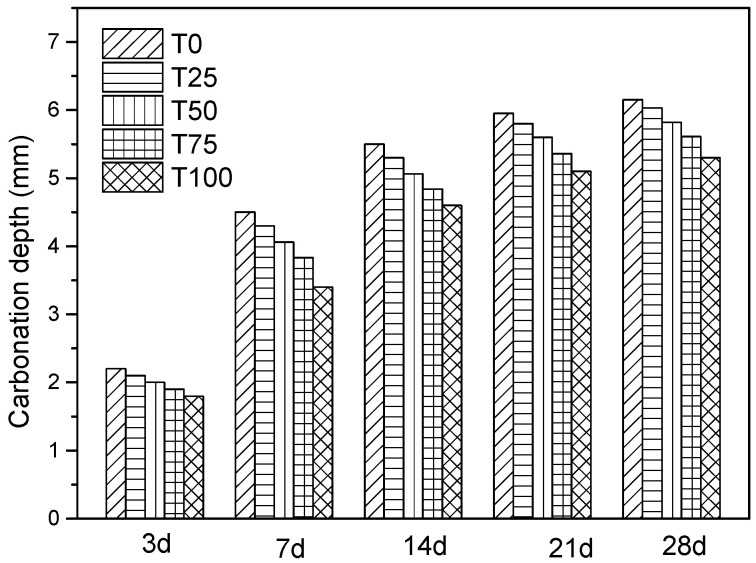
Comparison of carbonation depth.

**Table 1 materials-14-05606-t001:** The main chemical composition of cement (%).

CaO	SiO_2_	Al_2_O_3_	Fe_2_O_3_	MgO	SO_3_	Cl^−^	Loss
56.1	34.6	0.8	1.1	3.25	2.3	0.024	4.35

**Table 2 materials-14-05606-t002:** Mix ratio of the concrete (unit: kg/m^3^).

Description	ATCA Proportion				
	T0	T25	T50	T75	T100
Cement	485	485	485	485	485
NCA	1047	785.3	523.5	261.8	0
ATCA	0	261.8	523.5	785.3	1047
Manufactured sand	758	758	758	758	758
Water	160	160	160	160	160
Superplasticizer	5.8	5.8	5.8	5.8	5.8

**Table 3 materials-14-05606-t003:** The test results of fresh concrete.

Series	Slump (mm)		Air Content (%)		Setting Time (min)		Apparent Density (kg/m^3^)
	0 h	1 h	0 h	1 h	Initial	Final	
T0	198	187	3.8	2.9	398	570	2445
T25	195	183	3.7	2.8	389	565	2451
T50	192	182	3.7	2.7	380	556	2461
T75	188	179	3.4	2.7	372	547	2468
T100	185	175	3.3	2.6	365	540	2475

**Table 4 materials-14-05606-t004:** Dimensions of the concrete specimens for mechanical property experiments.

No.	Experiments	Sample Shape	Sample Size (mm)
1	Cubic compressive strength	Cube	150 × 150 × 150
2	Splitting tensile strength	Cube	150 × 150 × 150
3	Prism compressive strength	Prism	150 × 150 × 300
4	Elasticity modulus	Prism	150 × 150 × 300

**Table 5 materials-14-05606-t005:** Dimensions of the concrete specimens for shrinkage and creep.

No.	Experiments	Sample Shape	Sample Size (mm)
1	Shrinkage	Prism	100 × 100 × 515
2	Creep	Prism	100 × 100 × 400

**Table 6 materials-14-05606-t006:** Dimensions of the concrete specimens for durability performance experiments.

No.	Experiments	Sample Shape	Sample Size (mm)
1	Frost-resistant performance	Prism	100 × 100 × 400
2	Alkali activity	Prism	25 × 25 × 280
3	Chloride permeability	Cylinder	Ø100 × 50
4	Cracking sensitivity	Plate	800 × 600 × 100
5	Carbonation	Cube	100 × 100 × 100

## Data Availability

The data presented in this study are available on request from the corresponding author.

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
