# Peer review of "Strength and Durability Properties of Antimony Tailing Coarse Aggregate (ATCA) Concrete"

_materials, 2021, doi:10.3390/ma14195606_

Round 1

Reviewer 1 Report

Please see the attached. I recommend for publication and corresponding suggestions are in the word file. 

Author Response

We first would like to express our sincere appreciation to the Editor-in-Chief and Associate Editor for their time and efforts in coordinating the peer-review process, and four anonymous reviewers for their valuable comments on our manuscript. Their valuable comments significantly helped the authors to improve the quality of this manuscript. All the reviewers’ comments have been seriously taken into consideration and thoroughly addressed in the revised manuscript, and the explanation/corrections are provided in the following item-to-item response to each of the reviewers’ comments. To make the revised manuscript easier for editors and reviewers to read, all changes are printed in blue in the revised manuscript. Please see our replies to the reviewers’ comments for more details.

Reviewer 2 Report

The paper presents an experimental study carried out the feasibility of utilizing antimony tailing coarse aggregate (ATCA)13 in high strength concrete as a complete substitute with 25%, 50%, 75%, and 100% ATCA replacing the NCA. The paper is interesting and well written, and it provides helpful information. Although the paper has an acceptable level in terms of English writing, the paper should be improved considering the suggestions below.

  1. In Line 16, 22, and 350, we find that, we demonstrate, we have …. It should be appropriately rewritten.

Suggestions:  "the investigators," or "the researchers" instead of we.

  1. Chinese Standard GB/T 50080-2016 was employed for slump, air content, setting time and apparent141 density As a suggestion, it can be compared with other Standards like ACI and Eurocode since the results will be presented in international journals.

Author Response

(The authors gave the same response as above.)

Reviewer 3 Report

I like very much the structure of the article, research  plan, presentation of the research results. Very good article. Congratulations to the authors.

However, I have guestions:

  1. Whether the antimony waste rock is toxic? What is the chemical composition of the antimony tailing coarse aggregate?
  2. Have the authors tested the antimony tailing coarse aggregate? What is the strength of the antimony tailing coarse aggregate? This information would in the interpretation of the obtained test results.

Author Response

(The authors gave the same response as above.)

Reviewer 4 Report

The reviewer recommends reconsidering after the author conducts a major revision on the manuscript.

  1. The reference for table 1 should be added.
  2. Images of all actual experiments (compression, tensile, shrinkage, creep, …) that the authors conducted should be included in the manuscript.
  3. Base on the mix ratio of the concrete in table 2, we can see that the T0 sample will be a medium-strength concrete sample. However, the results presented in Figure 2 with compressive strength of all samples at time 28d is all higher than 50 MPa (super-high-strength concrete). Do the authors have any comments?
  4. As antimony (Sb) is a toxic element and its compounds are harmful substances, what do the authors think about the applicability of the ATCA concrete in civil structures?

Author Response

(The authors gave the same response as above.)

Round 2

Reviewer 4 Report

The authors have addressed all comments properly, and the revised manuscript is good enough for publication in the present form.